# A Comparative Study on Photo-Protective and Anti-Melanogenic Properties of Different *Kadsura coccinea* Extracts

**DOI:** 10.3390/plants10081633

**Published:** 2021-08-09

**Authors:** Joong Suk Jeon, He Mi Kang, Ju Ha Park, Jum Soon Kang, Yong Jae Lee, Young Hoon Park, Byoung Il Je, Sun Young Park, Young Whan Choi

**Affiliations:** 1Department of Horticultural Bioscience, Pusan National University, Myrang 627-706, Korea; jjhjswoll@naver.com (J.S.J.); mimi2965@naver.com (H.M.K.); juha31park@gmail.com (J.H.P.); kangjs@pusan.ac.kr (J.S.K.); yjl@pusan.ac.kr (Y.J.L.); ypark@pusan.ac.kr (Y.H.P.); bije@pusan.ac.kr (B.I.J.); 2Bio-IT Fusion Technology Research Institute, Pusan National University, Busan 609-735, Korea

**Keywords:** *Kadsura coccinea*, photoprotection, keratinocyte, anti-melanogenesis, melanocyte

## Abstract

*Kadsura coccinea* (KC), a beneficial plant for human health, has been used for centuries in China, Thailand, and Korea in folk medicine and food. There is evidence supporting the biological effects of highly bioactive ingredients in KC such as lignans, triterpenoids, flavonoids, phenolic acids, steroids, and amino acids. In this study, we aimed to explore the effects, functions, and mechanisms of the extracts from KC root (KCR), stem (KCS), leaf (KCL), and fruit (KCF) in UVA and UVB-irradiated keratinocytes and α-melanocyte stimulating hormone (α-MSH)-stimulated melanocytes. First, the total polyphenol and flavonoid contents of KCR, KCS, KCL, and KCF and their radical scavenging activities were investigated. These parameters were found to be in the following order: KCL > KCR > KCS > KCF. UVA and UVB-irradiated keratinocytes were treated with KCR, KCS, KCL, and KCF, and keratinocyte viability, LDH release, intracellular ROS production, and apoptosis were examined. Our results demonstrated that KC extracts improved keratinocyte viability and reduced LDH release, intracellular ROS production, and apoptosis in the presence UVA and UVB irradiation. The overall photoprotective activity of the KC extracts was confirmed in the following order: KCL > KCR > KCS > KCF. Moreover, KC extracts significantly decreased the intracellular melanin content and tyrosinase activity in α-MSH-stimulated melanocytes. Mechanistically, KC extracts reduced the protein and mRNA expression levels of tyrosinase, tyrosinase-related protein-1 (TRP-1), and tyrosinase-related protein-2 (TRP-2) in α-MSH-stimulated melanocytes. In addition, these extracts markedly downregulated myophthalmosis-related transcription factor expression and cAMP-related binding protein phosphorylation, which is upstream of the regulation of Tyrosinase, TRP-1, and TRP-2. The overall anti-melanogenic activity of the KC extracts was established in the following order. KCL > KCR > KCS > KCF. Overall, the KC extracts exert photoprotective and anti-melanogenic effects, providing a basis for developing potential skin-whitening and photoprotective agents.

## 1. Introduction

*Kadsura coccinea* (Lem.) B.C. Sm, also known as “black tiger” in China, is a species belonging to the economically and medically important family *Schisandraceae*. It is mainly cultivated in southern China, Thailand, and South Korea. *Kadsura coccinea* (KC) has also gained interest in Chinese folk medicine to identify effective treatments for preventing several diseases [1,2]. It is not only consumed as food but also is highly regarded for its pharmacological properties, especially anti-HIV, anti-fungal, anti-lipid peroxidation, anti-hepatitis, anti-inflammatory, and anti-tumor properties [3,4]. Many studies have shown the therapeutic effects of KC such as in treating gastrointestinal disorders and rheumatoid arthritis, calming the heart, strengthening the kidneys, and promoting blood and fluid circulation [5,6]. It is a novel and rare species with valuable root, stem, leaf, and fruit parts used in traditional Dai medicines (TDM). Many flavonoids and phenolic acids have been found in high concentrations in KC extracts, and these compounds are believed to contribute to the medicinal properties of KC extracts [6,7]. Given the pharmacological properties of KC, the efficacy of extracts from different parts of KC with regard to photoprotective and anti-melanogenic properties is worth exploring.

Solar ultraviolet (UV) radiation, characterized by UVA (320–400 nm), UVB (280–320 nm), and UVC (100–280 nm), is the most critical environmental factor causing skin cancer and photoaging resulting from cytotoxicity, genotoxicity, and phototoxicity. Specifically, UVA and UVB radiation comprise over 95% and 3% of daily UV irradiation, respectively [8]. UVA and UVB irradiation can induce reactive oxygen species (ROS) indirectly or directly by penetrating the epidermal and/or dermal layers of the skin, leading to oxidative damage and cell death. In recent decades, UV radiation has become a serious skin health concern and is still spreading dangerously worldwide [9,10,11]. UV irradiation is a direct and consistent stimulant of keratinocytes, which account for approximately 95% of the skin epidermal cell mass. Keratinocytes act as the first barrier against microbial, chemical, and physical hazards, and helps defend against UVA and UVB radiation. When keratinocytes are exposed to UVA and UVB, intracellular ROS are generated, thus triggering apoptosis [12,13,14].

Melanocytes are responsible for the production and quantity of melanin pigments, which are important players in the biological defense of the skin epidermis; their dysregulation can cause hyperpigmentation or hypopigmentation disorders [15,16]. Melanocytes are distributed in the stratum basale of the skin epidermis, which is also affected by sunlight exposure, reactive oxygen species (ROS), and α-melanocyte stimulating hormones (α-MSH) [17,18]. Tyrosinase, a member of the type-3 copper protein family, is an evolutionarily conserved metalloprotein that plays a crucial role in melanogenesis and in monophenol monooxygenase, catecholase, and diphenolase activities. The downregulation of tyrosinase, tyrosinase-related protein-1 (TRP-1), and tyrosinase-related protein-2 (TRP-2) has distinct catalytic effects. TRP-1 is a 5,6-dihydroxyindole-2-carboxylic acid oxidase, and TRP-2 is a DOPAchrome tautomerase [19,20]. In addition, the myophthalmosis-related transcription factor (MITF) and cAMP responsive element binding protein (CREB) are transcription factors that primarily regulate melanogenesis and encode information about the mode and intensity of stimulation [17,21]. Multiple studies have stated that MITF and CREB pathways regulate melanogenesis. MITF is a key factor in the transcription of melanogenesis-related enzymes and the central regulator of melanogenesis. α-MSH leads to the expression of MITF through a signaling mechanism involving cAMP-related binding protein (CREB). Then, MITF enters the nucleus with the M-box sequence (AGTCATGTGCT) to promote the transcription of specific melanogenic genes and enzymes. It is also known that phosphorylated CREB is stimulated by α-MSH, which binds to the CRE consensus element in the *Mitf* promoter to upregulate *Mitf* transcription [21,22,23,24].

In this study, extracts of KC root (KCR), stem (KCS), leaf (KCL), and fruit (KCF) were comprehensively compared and multiple evaluations of their photoprotective and anti-melanogenic properties were performed.

## 2. Materials and Methods

### 2.1. Preparation of KC Extracts 

A three-year-old KC 15 plant was grown in a 9 L plastic pot at Miryang campus of the Pusan National University. KC was identified by Professor Young Whan Choi, an author of this study. These samples were deposited as voucher specimens (accession number KC-PDRL-1) at the herbarium of the Department of Horticultural Bioscience, College of Natural Resources and Life Science, Pusan National University. Plants were watered sufficiently using a complete nutrient solution with a conductivity level of 1.0 mS·cm^−1^ and containing the following elements (in me∙L^−1^): NO_3_-N, 16; NH_4_-N, 1.34; P, 4; K, 8; Ca, 8; and S, 4. KC gown in the port was harvested in December 2020 by classifying roots, stems, leaves, and fruits (Figure 1A). Harvested samples were immediately lyophilized in a freeze dryer and stored in vinyl bags at −20 °C until analysis. The dried roots, stems, leaves, and fruits of KC (20 g) were ground to a fine powder and extracted at room temperature with ethyl alcohol. Briefly, filtration and evaporation of EtOH extracts of KC were performed under reduced pressure at 45 °C followed by lyophilization. Finally, the solid extract (50 mg/mL) was dissolved in dimethyl sulfoxide (DMSO) for further experiments.

### 2.2. Total Polyphenolic and Flavonoid Contents of KC Extracts

As described previously [25], the total polyphenolic and flavonoid contents of KC root, stem, leaf, and fruit extracts were measured using the Folin–Ciocalteu (total polyphenol) and aluminum chloride (flavonoid) colorimetric methods. Absorbance was measured at 700 nm (total polyphenol) using Ultrospec 6300 Pro (GE Healthcare Life Sciences, Buckinghamshire, UK) and at 510 nm (flavonoid) using the VICTOR Multilabel Plate Reader (Perkin-Elmer, Waltham, MA, USA).

Standard curves were constructed using gallic acid (total polyphenol) and quercetin (flavonoid) as the standards, and the results were expressed as gallic acid equivalents per gram (GAE/g) of KC root, stem, leaf, and fruit extracts and quercetin equivalent per gram (QE/g) of KC root, stem, leaf, and fruit extracts, respectively.

### 2.3. DPPH and ABTS Assay

The DPPH and ABTS radical scavenging activities of KC root, stem, leaf, and fruit extracts (0.5 mg/mL) were measured according to a previously described method [25] with slight modifications. KC root, stem, leaf, and fruit extracts (0.5 mg/mL) were mixed with DPPH solution (60 µM) in microplates. After the samples were shaken vigorously, they were kept in the dark at 25 °C for 0.5 h. Stock solutions of 7 mM ABTS and 2.6 mM potassium persulfate were prepared in distilled water at room temperature in the dark for 18 hrs. KC root, stem, leaf, and fruit extracts (0.5 mg/mL) were mixed with the working solution and then allowed to stand for 0.5 h at room temperature in the dark. The absorbance of the sample mixtures was monitored at 510 nm (DPPH) or 734 nm (ABTS).

### 2.4. Cell Culture

The adhered Keratinocytes (HaCaT) and adhered melanocytes (B16F10) were inoculated in DMEM (Invitrogen Corporation, Carlsbad, CA, USA) culture solution containing 10% FBS (Invitrogen Corporation, Carlsbad, CA, USA) and 1% penicillin/streptomycin (Invitrogen Corporation, Carlsbad, CA, USA) and cultured at 37 °C with 5% CO_2_. The adhered HaCaT and B16F10 cell growth was observed regularly, and the medium was changed every two to three days. Cells from the logarithmic growth phase were used for subsequent experiments.

### 2.5. UVA and UVB Irradiation

Keratinocytes were exposed to UVA or UVB irradiation (Bio-Link BLX-365, Villber-Lourmat, Eberhardzell, Germany) with 5 × 8 W tubes that emit most of their energy at an emission peak at either 365 nm (UVA) or 312 nm (UVB). UVA irradiation doses were 20 J/cm^2^ and UVB irradiation doses were 50 mJ/cm^2^.

### 2.6. Measurement of Cell Viability and Cytotoxicity through CCK-8 and LDH Analysis

For cell viability analysis, Cell Counting Kit-8 (CCK-8) solution from the CCK-8 Assay Kit (Sigma-Aldrich, St. Louis, MO, USA) was added to HaCaT keratinocytes and B16F10 melanoma cell suspensions according to the manufacturer’s instructions and incubated at 37 °C for 4 h. Briefly, 2 × 104 cells were seeded into each well of a 24-well plate and incubated with 5% CO at 37 °C for 24 h. Briefly, 2 × 104 cells were seeded into each well of a 24-well plate and incubated with 5% CO at 37 °C for 24 h. After 24 h of incubation, CCK-8 reagent was added to each well, and the cells were further incubated for 4 h. A cytotoxicity detection kit (Roche Applied Science, Switzerland) was used to determine extracellular lactate dehydrogenase (LDH) release in HaCaT keratinocyte culture medium. Absorbance was analyzed at 450 nm (CCK-8) and 490 nm (LDH) using a VICTOR Multilabel Plate Reader.

### 2.7. Evaluation of Intracellular ROS Production in Keratinocytes

Intracellular ROS levels were analyzed using 5-(and-6)-chloromethyl-2′,7′-dichlorodihydrofluorescein diacetate acetyl ester (CM-H_2_DCFDA; Thermo Fisher Scientific, Waltham, MA, USA). All procedures were performed according to the manufacturer’s instructions. In brief, after KC other partial extracts (0.5 mg/mL) treatment, Keratinocytes (HaCaT) were washed with phosphate-buffered saline (PBS) and incubated with CM-H_2_DCFDA (5 µM) for 0.5 h in the dark. Thereafter, the generation of intracellular ROS was visualized using a Carl Zeiss fluorescence microscope; fluorescence intensity was measured based on a fluorescent dye (CM-H_2_DCFDA) using a flow cytometer (Fit NXT Flow Cyto, Thermo Fisher Scientific, Pasadena, CA, USA).

### 2.8. Analysis of Apoptosis

After exposure and treatment, HaCaT keratinocytes were trypsinized and centrifuged. Subsequently, apoptosis of the obtained cells was evaluated using the fluorescein isothiocyanate (FITC) Annexin V/Dead Cell Apoptosis Kit (Invitrogen Life Technologies, Carlsbad, CA, USA) according to the manufacturer’s instructions. Briefly, keratinocytes were rinsed twice with PBS and keratinocytes in the annexin binding buffer were obtained and mixed with FITC/Annexin V (component A) and propidium iodide (PI) working solution. After incubation at room temperature for 15 min in the dark, keratinocyte apoptosis was measured and the percentage of apoptotic cells were calculated using a flow cytometer (Fit NXT Flow Cyto, Thermo Fisher Scientific, Pasadena, CA, USA). Signals were detected for FL1 (FITC/Nexin V) and FL3 (PI) channels, and quadrant marker staining and dot plots of the stained cells were established.

### 2.9. Analysis of Intracellular Melanin Content and Tyrosinase Activity

Intracellular melanin content and tyrosinase activity assays were measured by explaining where the slightly modified procedure differed [24]. Melanocytes were treated with final concentration of 0.5 μM α-MSH and the 0.5 mg/mL KC other partial extracts for 48 h. Melanocyte pellets were lysed with 1 N NaOH in 10% DMSO at 80 °C for 1 h. Relative melanin content was determined by measuring absorbance at 475 nm using a VICTOR Multilabel Plate Reader. Intracellular tyrosinase activity was determined by measuring the rate of dopachrome production using L-DOPA. Melanocytes were washed with ice-cold PBS and lysed in PBS containing 1% (*w*/*v*) Triton X-100. The tyrosinase substrate L-DOPA (2 mg/mL) was prepared in the same phosphate lysis buffer. Each extract was placed in a 96-well plate, and enzyme analysis was initiated by adding L-DOPA solution. After incubation for 1 h, the absorbance was measured at 475 nm using a VICTOR Multilabel Plate Reader to analyze dopachrome production. The value of each measurement was expressed as a percentage of the control. Arbutin (A, 0.5 mg/mL) was used as a positive control.

### 2.10. Quantitative Real-Time PCR

Total RNA was isolated from each group of melanocytes using the RNeasy Mini Kit (QIAGEN, Hilden, Germany). Reverse transcription was performed using the high-capacity cDNA reverse transcription kit (Thermo Fisher Scientific, Miami, OK, USA), according to the manufacturer’s instructions, to obtain the first strand of cDNA. The strands were then used as templates for quantitative real-time PCR (qRT-PCR) using a Bio-Rad Chromo4TM instrument and SYBR Green qPCR master mix (Thermo Fisher Scientific, Miami, OK, USA). PCR was performed under pre-denaturation at 95 °C for 5 min, denaturation at 95 °C for 15 s, and annealing at 55–58 °C for 30 s. GAPDH mRNA was used as an internal reference for tyrosinase, TRP-1, and TRP-2 mRNA. Relative value of target gene expression = 2^−∆∆CT^. Primer sequences were as follows: Tyrosinase-sense (5′-ggccagctttcaggcagaggt-3′), Tyrosinase-anti-sense (5′-tggtgcttcatgggc aaaatc-3′), TRP-1-sense (5′-agccccaactctgtcttttc-3′), TRP-1-anti-sense (5′-ggtctccctacatttccagc-3′), TRP-2-sense (5′-tccagaagtttgacagccc-3′), TRP-2-anti-sense (5′-ggaaggagtgagccaagttatg-3′), GAPDH-sense (5′-aggtggtctcctctgacttc-3′), and GAPDH-anti-sense (5′-taccaggaaatgagcttgac-3′).

### 2.11. Western Blotting 

Melanocytes were harvested and lysed using mammalian protein extraction reagent (Thermo Fisher Scientific, Waltham, MA, USA). All procedures were performed according to the manufacturer’s instructions. All protein concentrations were determined using a Bio-Rad protein assay kit (Bio-Rad, Hercules, CA, USA). Then, loading buffer was added to the protein supernatant and mixed. The mixture was boiled for 10 min, and the proteins were separated using Mini-PROTEAN Precast Gels (Bio-Rad, Hercules, CA, USA) and transferred onto a Hybond polyvinylidene difluoride membrane (Amersham Biosciences, Piscataway, NJ, USA). Immunodetection was performed using tyrosinase (1:1000), TRP-1 (1:1000), TRP-2 (1:1000), MITF (1:1000), phosphorylated CREB (p-CREB 1:1000), CREB (1:1000), and α-tubulin (1:1000) (Cell Signaling Technology, Beverly, MA, USA) using the SignalBoost Immunoreaction Enhancer Kit (Sigma-Aldrich, St. Louis, MO, USA). The membrane was incubated overnight with the primary antibodies at 4 °C. The goat anti-rabbit (IgG) secondary antibody (1:5000, Cell Signaling Technology) was added to the membrane and incubated at room temperature for 1 h. Protein bands were observed using an enhanced Pierce ECL western blotting Substrate (Thermo Fisher Scientific, Waltham, MA, USA) and quantified as the ratio of the intensity of the target protein band to the intensity of the α-tubulin band.

### 2.12. Statistical Analysis

All assays were independently repeated at least three times. All statistical parameters are presented as the mean ± standard error of the mean (SEM). Statistical analyses were performed using one-way analysis of variance (ANOVA), followed by Dunn’s post-hoc test. A value of *p* < 0.01 or *p* < 0.05 was considered significant.

## 3. Results

### 3.1. Comparison of Antioxidant Properties of Several Partial Extracts of KC

The total polyphenol and flavonoid contents and ABTS and DPPH scavenging activities were compared to determine the potential effects of KCR, KCS, KCL, and KCF extracts on antioxidant capacity. As shown in Figure 1B, the KCR extract (197.6 ± 27.2 mg GAE/g) exhibited the highest phenol content, followed by KCL (153.7 ± 6.7 mg GAE/g) and KCS (88.1 ± 7.8 mg GAE/g); the KCF (21.8 ± 4.8 mg GAE/g) extract had the lowest phenol content. Moreover, the results of flavonoid content analyses showed that KCL (94.5 ± 6.3 mg QE/g) had the highest flavonoid content followed by KCR (79.6 ± 4.2 mg QE/g); the KCS (14.9 ± 1.3 mg QE/g) and KCF (6.4 ± 2.0 mg QE/g) extracts had the lowest flavonoid content (Figure 1C). To further investigate the antioxidant properties of KCR, KCS, KCL, and KCF, ABTS and DPPH radical scavenging assays were performed. As shown in Figure 1D, KCL (94.7 ± 2.9%) and KCR (82.8 ± 5.9%) exhibited the highest ABTS radical scavenging activity, followed by the KCS (29.7 ± 2.0%) and KCF extract (15.9 ± 2.0%). DPPH radical scavenging activity was also shown in the order of KCL (99.9 ± 0.1%) > KCR (95.5 ± 3.6%) > KCS (25.7 ± 2.1%) > KCF (8.7 ± 1.1%) (Figure 1E).

### 3.2. Comparison of Viable and Damaged Keratinocytes Treated with Several Partial Extracts of KC in the Presence of UVA, UVB, or Non-Irradiation

We conducted the following experiments to explore the effects of KCR, KCS, KCL, and KCF on keratinocytes. First, all of the extracts were applied to HaCaT keratinocytes in the presence of UVA, UVB, or non-irradiation. CCK-8 analysis showed that the extracts did not significantly change keratinocyte viability at concentrations of 0.5 mg/mL. Later, keratinocytes were treated with KCR, KCS, KCL, and KCF in the presence UVA or UVB irradiation. UVA and UVB irradiation significantly inhibited keratinocyte viability, as shown by CCK-8 analysis in Figure 2A. However, we found that the viability of UVA- and UVB-irradiated keratinocytes increased in the following order: KCL, KCR, KCS, and KCF (Figure 2A). We also monitored the damaged keratinocytes by measuring the extracellular LDH release. The results showed that UVA or UVB irradiation significantly facilitated LDH release; however, KCL, KCR, KCS, and KCF significantly attenuated LDH release with UVA or UVB exposure in that order (Figure 2B). The above experimental results showed that the anti-proliferative and cytotoxic effects of UVA or UVB irradiation on keratinocytes were alleviated in the order of KCL, KCR, KCS, and KCF. Notably, KCL can restore cell viability to control levels.

### 3.3. Comparison of Intracellular ROS Production in Keratinocytes Treated with Several Partial Extracts of KC in the Presence or Absence of UVA and UVB Irradiation

As intracellular production of ROS causes severe keratinocyte damage, they are considered potential mediators of UVA or UVB radiation-induced damage. Several studies have suggested that UVA or UVB irradiation induces endogenous ROS production [12,14]. We aimed to determine if there was an increase in endogenous ROS levels in UVA-or UVB-irradiated keratinocytes. Accordingly, we analyzed the intracellular fluorescence intensity of the probe CM-H_2_DCFDA using fluorescence microscopy and flow cytometry. As a result of fluorescence microscopy, CM-H_2_DCFDA staining images showed slight staining in control, KCR, KCS, KCL, and KCF-treated keratinocytes and significant staining in UVA-or UVB-irradiated keratinocytes (Figure 3A). According to the quantified results of flow cytometry, UVA and UVB irradiation increased intracellular ROS levels in keratinocytes by 27.2 ± 4.5% and 34.1 ± 4.2%, respectively, compared with that in the control (5.7 ± 0.2%). In addition, similar to the fluorescence microscopy results, it was confirmed that the intracellular ROS level was suppressed by KC extracts in the order of KCL > KCR > KCS > KCF in the presence UVA or UVB irradiation (Figure 3B). These results indicate that several partial extracts of KC significantly inhibited keratinocyte damage by reducing endogenous ROS levels.

### 3.4. Comparison of Apoptosis of Keratinocytes Treated with Several Partial Extracts of KC in the Presence or Absence of UVA and UVB Irradiation

To detect apoptosis, which is a reliable indicator of keratinocyte damage, keratinocytes were stained with Annexin V in combination with propidium iodide. The fluorescein isothiocyanate (FITC) Annexin V/Dead Cell Apoptosis Kit was used to test the rate of apoptosis in keratinocytes. UVA and UVB irradiation facilitated the Annexin V staining activity, whereas KCR, KCS, KCL, and KCF decreased the Annexin V staining activity rate in the presence UVA and UVB irradiation. KCR, KCS, KCL, and KCF alone (0.5 mg/mL) did not induce Annexin V staining activity. Quantified data from flow cytometry showed that UVA and UVB irradiation increased apoptosis levels of keratinocytes by 46.3 ± 1.5% and 48.7 ± 1.0%, respectively, compared with that of the control (5.0% ± 0.7%). Importantly, it was confirmed that the level of apoptosis was suppressed by KC extracts in the order of KCL > KCR > KCS > KCF in the presence of UVA or UVB irradiation (Figure 4A,B). Overall, UVA and UVB irradiation caused keratinocyte damage and several partial extracts of KC attenuated keratinocyte damage induced by UV irradiation.

### 3.5. Comparison of Intracellular Melanin Content and Tyrosinase Activity of Melanocytes Treated with Several Partial Extracts of KC in the Presence or Absence of α-MSH Treatment

Prior to investigating the biological potential of KCR, KCS, KCL, and KCF on α- MSH-induced melanogenesis, the cell viability after KCR, KCS, KCL, and KCF (0.5 mg/mL) treatment was assessed using the CCK-8 assay in melanocytes B16F10 with or without α-MSH. KCR, KCS, KCL, and KCF (0.5 mg/mL) did not alter cell viability in the presence or absence of α-MSH (Figure 5A). α-MSH is an important melanogenic agent that can increase intracellular melanin content by binding to the melanocortin 1 receptor and activating adenylate cyclase. To investigate the effect of KCR, KCS, KCL, and KCF on melanogenesis in melanocytes, the intracellular melanin content was determined by visual observation and biochemical measurements. As shown in Figure 5B, the intracellular melanin content was significantly increased by α-MSH. However, co-treatment with KCR, KCS, KCL, and KCF showed a remarkable reduction in the intracellular melanin content compared with the α-MSH treatment. The sequence of biochemical measurements indicating the inhibition of intracellular melanin content was as follows: KCL > KCR > KCS > KCF (Figure 5B). Intracellular tyrosinase activity assay was performed according to the intracellular melanin content assay. The intracellular tyrosinase activity of α-MSH-stimulated melanocytes increased, whereas that of α-MSH-stimulated melanocytes treated with KCR, KCS, KCL, and KCF decreased. The sequence of biochemical measurements indicating the inhibition of intracellular tyrosinase activity was as follows: KCL > KCR > KCS > KCF (Figure 5C). These results indicate that several partial extracts of KC significantly reduced intracellular melanin content and inhibited tyrosinase activity without altering cell viability.

### 3.6. Comparison of Transcription and Translation Levels of Tyrosinase, TRP-1, and TRP-2 in Melanocytes Treated with Several Partial Extracts of KC in the Presence or Absence of α-MSH Treatment

To explore the effect of KCR, KCS, KCL, and KCF on downregulation of the protein and mRNA expression levels of melanogenesis markers (tyrosinase, TRP-1, and TRP-2), the melanocytes were treated with KCR, KCS, KCL, and KCF in the presence or absence of α-MSH. As shown in Figure 6A–C, the mRNA levels of tyrosinase, TRP-1, and TRP-2 were significantly increased by α-MSH treatment. In contrast, compared with the α-MSH treatment, KCR, KCS, KCL, and KCF downregulated the mRNA expression of tyrosinase, TRP-1, and TRP-2. In addition, KCR, KCS, KCL, and KCF alone had no significant effect on tyrosinase, TRP-1, and TRP-2 mRNA levels. Western blot analysis was performed in cooperation with real-time PCR. The results showed that α-MSH treatment increased the protein expression levels of tyrosinase, TRP-1, and TRP-2 and these levels were decreased by co-treatment with KCR, KCS, KCL, and KCF (Figure 6D). The sequence of quantitative real-time PCR and western blot indicating inhibition of tyrosinase, TRP-1, and TRP-2 mRNA and protein expression was as follows: KCL > KCR = KCS > KCF (Figure 6). These results suggest that several partial extracts of KC have the potential to promote anti-melanogenesis, which was shown by the downregulation of melanogenesis markers. 

### 3.7. Comparison of MITF Expression and CREB Phosphorylation of Melanocytes Treated with Several Partial Extracts of KC in the Presence or Absence of α-MSH Treatment

Tyrosinase, TRP-1, and TRP-2 are essential in melanogenesis. Their expression is regulated by MITF expression and CREB phosphorylation [17,21]. Western blot analysis indicated that KCR, KCS, KCL, and KCF effectively inhibited the increase in the protein level of MITF caused by α-MSH treatment. In addition, KCR, KCS, KCL, and KCF alone hardly detected MITF protein expression. The sequence of quantified western blot results indicating the inhibition of MITF protein expression levels was as follows: KCL > KCR > KCS > KCF (Figure 7A). Moreover, KCR, KCS, KCL, and KCF reversed the effects of α-MSH treatment on CREB phosphorylation. KCR, KCS, KCL, and KCF alone had little effect on CREB phosphorylation. The sequence of quantified western blot results indicating inhibition of CREB phosphorylation levels was as follows: KCL > KCR > KCS > KCF (Figure 7B). These results indicated that several partial extracts of KC suppressed melanogenesis in melanocytes, at least in part, through MITF expression and CREB phosphorylation.

## 4. Discussion

The popularity of skin whitening is increasing worldwide owing to the increase in UV irradiation, and it is likely to reach high proportions in the coming decades for esthetic purposes [8]. The numerous types of bleach, such as kojic acid and arbutin, have been used in the cosmetic and pharmaceutical markets. Moreover, natural extracts are receiving increasing attention owing to their potential antioxidant, anti-inflammatory, anti-tumor, antibacterial, and other activities [26,27]. Based on the characteristics of antioxidants and photoprotective and anti-melanogenesis, several cosmetic and pharmaceutical candidates have been developed. It is well known that these properties of whitening candidates are indispensable contributors to cosmetic and pharmaceutical research and development [28]. TDM has multi-component and multi-target properties and greatly improves human biological effectiveness and quality of life [29]. Modern phytochemical research shows that KC contains a variety of ingredients, with lignans and terpenoids being the main ingredients [30]. More than 202 compounds have been identified, including dibenzocyclooctadiene lignans, Spirobenzofuranoid dibenzocyclooctadiene lignans, Arylnaphthalene lignans, Kadlongilactone triterpenoids, and sesquiterpenoids [31,32]. The dried roots, stems, and leaves of KC have a broad tradition of use in TDM to treat rheumatoid arthritis, duodenal ulcers, gastrointestinal disorders, and gynecological problems. The dried roots of KC, with actions of clearing heat and eliminating toxins, inducing diuresis for removing edema. Fruits of KC are mostly consumed in the form of fresh fruits, juices, and fruit wine, indicating that they are beneficial to human health [7,33,34]. Previous studies also have shown that it is rich in bioactive ingredients such as lignans, triterpenoids, flavonoids, phenolic acids, steroids, and amino acids, which have high nutritional and medicinal value [3,35]. In this study, different parts of KC were extracted. Notably, the total polyphenol and flavonoid contents of KC leaves and roots were much higher than those of stems and fruits. Leaves and roots contain more than twice as many polyphenols and flavonoids as stems and fruits. As a result, we consider that total polyphenols and flavonoids make a significant contribution to the photoprotective and anti-melanogenic effects of KC.

Cutaneous phototoxicity caused by UV radiation is primarily due to cell cytotoxicity, intracellular ROS accumulation, and apoptosis in keratinocytes. Therefore, it is more reasonable to focus on the inhibition of cell cytotoxicity, intracellular ROS accumulation, and apoptosis in UVA-and UVB-irradiated keratinocytes [36,37]. As described in this study [10,38], the intervention with KC extracts on photo-cytotoxicity (UVA: 20 J/cm^2^, UVB: 50 mJ/cm^2^) results showed that KC extracts significantly reduced the cell cytotoxicity, intracellular ROS accumulation and apoptosis. Consistent with previous results, the photo-protective effects of KC leaves and roots were much higher than those of the stem and fruit. Furthermore, our results indicated that KC extracts show the above-mentioned effects by promoting antioxidant activity. Melanogenesis is often observed after UV irradiation and is primarily associated with pigmentation or hyperpigmentation [39]. According to previous studies, the method of inducing melanogenesis using α-MSH has been widely recognized and applied. Therefore, this study was based on the construction of a melanocyte model stimulated with α-MSH [24,39]. We found that KC extracts suppressed α-MSH-stimulated intracellular melanin content and tyrosinase activity. In addition, KC extracts downregulated the transcription and translation of melanogenesis markers such as tyrosinase, TRP-1, and TRP-2 in α-MSH-stimulated melanocytes. Furthermore, we investigated MITF protein expression and CREB phosphorylation in α-MSH-stimulated melanocytes. Similarly, in this study, we found that KC extracts suppressed α-MSH-mediated MITF protein expression and CREB phosphorylation in melanocytes. Consistent with photoprotective results, the anti-melanogenic effects of KC leaves and roots were much higher than those of the stems and fruits.

## 5. Conclusions

Overall, the potential photoprotective and anti-melanogenic effects of the KC extract were found in this study. KC extract with high polyphenol and flavonoid contents can exert photo-protective and anti-melanogenic effects on keratinocytes and melanocytes. This study provides a rationale and research strategy for cosmetic and pharmaceutical interventions for natural whitening and photoprotective agents.

## Figures and Tables

**Figure 1 plants-10-01633-f001:**
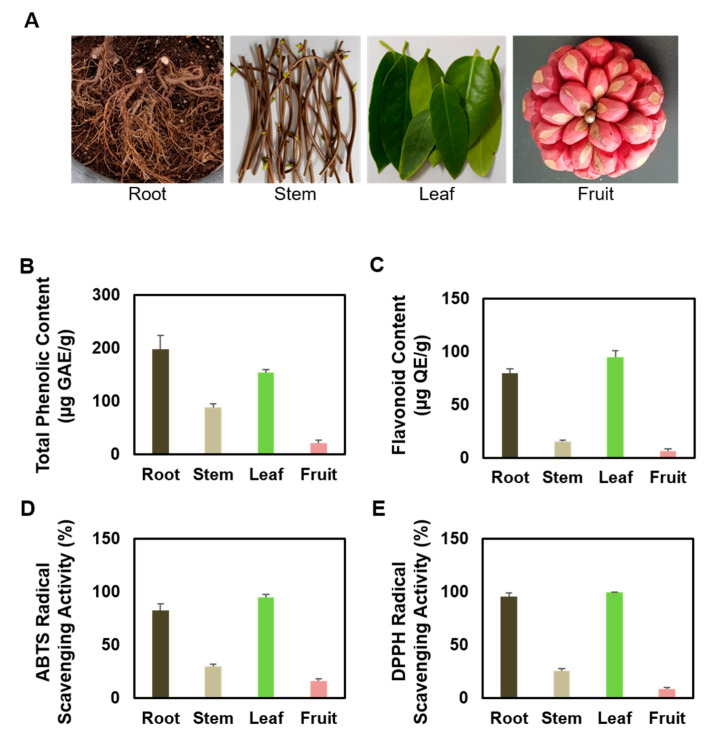
Antioxidant effects of KCR, KCS, KCL and KCF extract. (**A**) Photographs of KC. (KCR, root; KCS, stem; KCL, leaf; KCF, fruit). (**B**) Total polyphenolic content, (**C**) Flavonoid content, (**D**) ABTS radical scavenging activity, and (**E**) DPPH radical scavenging activity of KCR, KCS, KCL, and KCF extract (0.5 mg/mL). Data are expressed as mean ± SEM (*n* = 3). DMSO was used as a negative control.

**Figure 2 plants-10-01633-f002:**
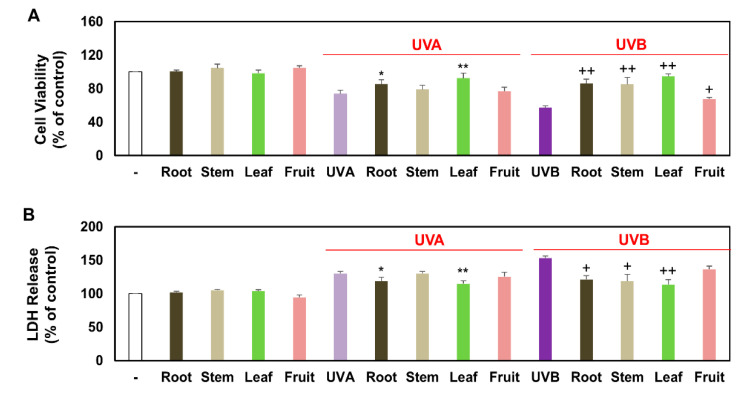
The keratinocytes viability and cytotoxicity effects of KCR, KCS, KCL, and KCF extract in the presence of UVA, UVB, or non-irradiation. Keratinocyte HaCaT were treated with KCR, KCS, KCL, and KCF extract (0.5 mg/mL) for 1 h after being exposed to UVA (20 J/cm^2^), UVB (50 mJ/cm^2^) or non-irradiation. These keratinocytes were maintained for 24 h. (**A**) The percentage of viable keratinocytes was assessed using the CCK-8 assay. (**B**) The percentage of cytotoxicity was measured using the LDH assay. * *p* < 0.05 and ** *p* < 0.01 (vs. UVA-treatment only). + *p* < 0.05 and ++ *p* < 0.01 (vs. UVB-treatment only). Data are expressed as mean ± SEM (*n* = 3).

**Figure 3 plants-10-01633-f003:**
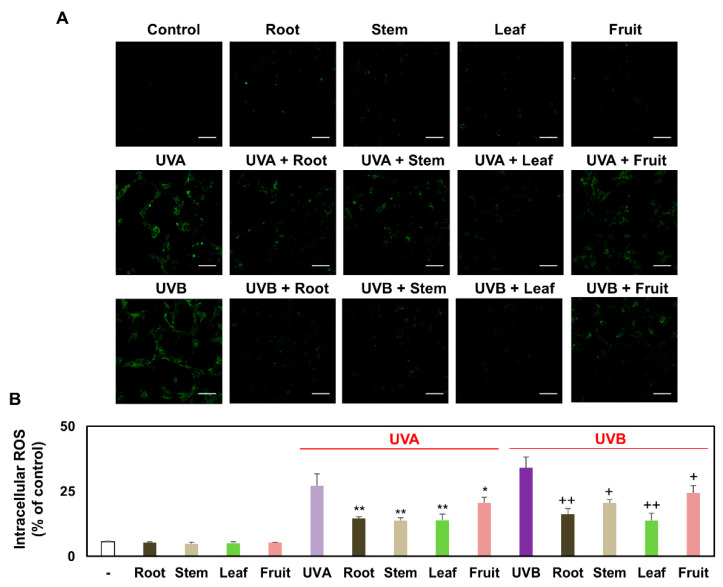
Effect of KCR, KCS, KCL, and KCF extract on the intracellular ROS production in UVA, UVB, or non-irradiated keratinocytes. Keratinocyte HaCaT were treated with KCR, KCS, KCL, and KCF extract (0.5 mg/mL) for 1 h after being exposed to UVA (20 J/cm^2^), UVB (50 mJ/cm^2^), or non-irradiation. These keratinocytes were maintained for 24 h. (**A**) Intracellular ROS production was visualized by Carl Zeiss fluorescence microscopy using CM-H_2_DCFDA staining (scale bar; 100 μm). (**B**) Representative flow cytometer-based quantitative analysis show fluorescence intensity using CM-H_2_DCFDA staining. * *p* < 0.05 and ** *p* < 0.01 (vs. UVA-irradiated group). + *p* < 0.05 and ++ *p* < 0.01 (vs. UVB-irradiated group). Data are expressed as mean ± SEM (*n* = 3).

**Figure 4 plants-10-01633-f004:**
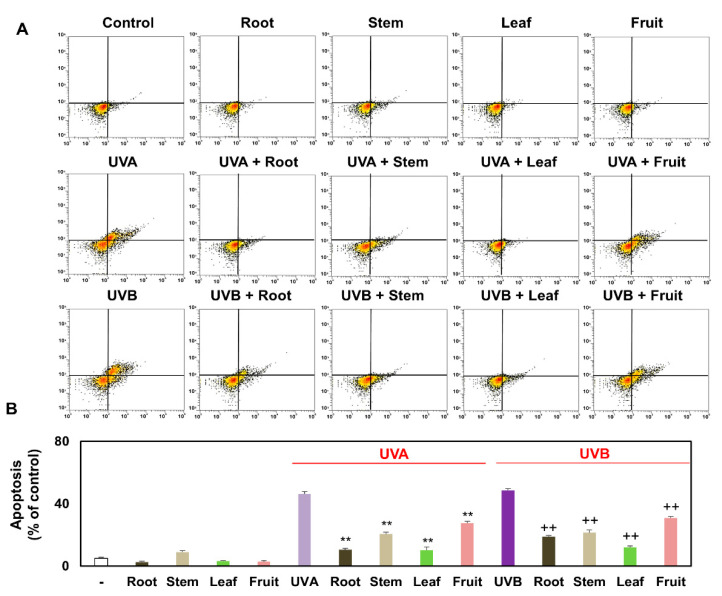
Effect of KCR, KCS, KCL, and KCF extracts on the apoptosis in UVA, UVB, or non-irradiated keratinocytes. Keratinocyte HaCaT were treated with KCR, KCS, KCL, and KCF extract (0.5 mg/mL) for 1 h after being exposed to UVA (20 J/cm^2^), UVB (50 mJ/cm^2^), or non-irradiation. These keratinocytes were maintained for 24 h. Representative flow cytometer-based images (**A**) and quantitative analysis (**B**) show fluorescence intensity using Annexin V/Dead Cell Apoptosis Kit. ** *p* < 0.01 (vs. UVA-irradiated group). ++ *p* < 0.01 (vs. UVB-irradiated group). Data are expressed as mean ± SEM (*n* = 3).

**Figure 5 plants-10-01633-f005:**
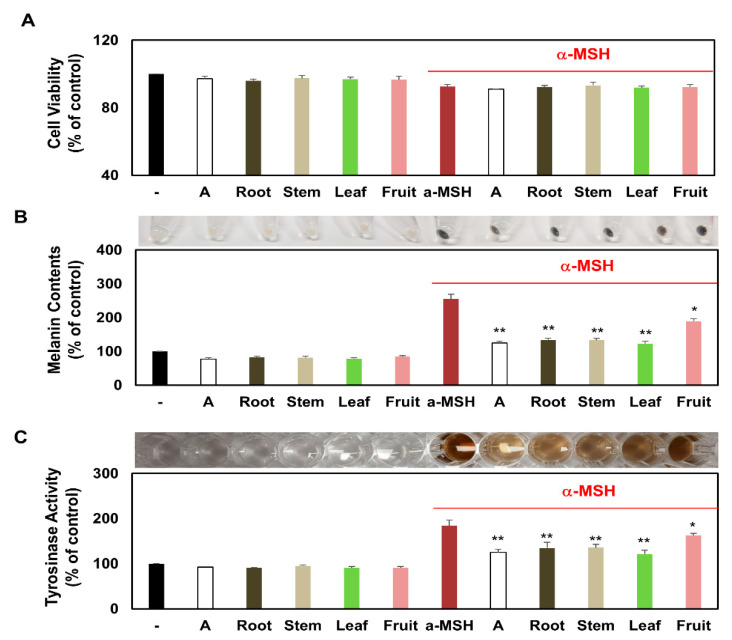
The melanin synthesis and tyrosinase activity of KCR, KCS, KCL, and KCF extract in α-MSH-stimulated melanocytes. Melanocytes B16F10 were treated with KCR, KCS, KCL, and KCF extract (0.5 mg/mL) for 1 h after being treated to α-MSH (0.5 μM) or non-treatment. These melanocytes were maintained for 48 h. (**A**) The percentage of viable melanocytes was assessed using the CCK-8 assay. Intracellular melanin content (**B**) and tyrosinase activity assays (**C**) were determined. Relative mean values were calculated by normalizing for protein content. * *p* < 0.05 and ** *p* < 0.01 (vs. α-MSH-stimulated group). Arbutin (A, 0.5 mg/mL) was used as a positive control. Data are expressed as mean ± SEM (*n* = 3).

**Figure 6 plants-10-01633-f006:**
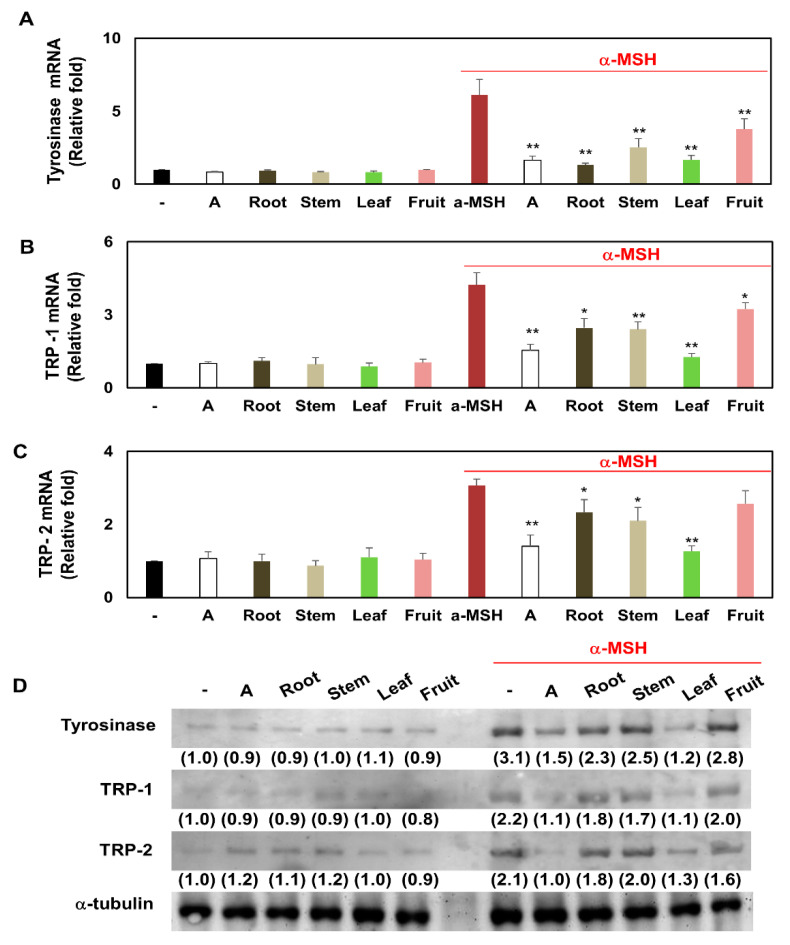
The mRNA and protein expression of tyrosinase, TRP-1, and TRP-2 of KCR, KCS, KCL, and KCF extract in α-MSH-stimulated melanocytes. Melanocytes B16F10 were treated with KCR, KCS, KCL, and KCF extract (0.5 mg/mL) for 1 h after being treated to α-MSH (0.5 μM) or non-treatment. These melanocytes were maintained for 48 h. (**A**) Tyrosinase, (**B**) TRP-1, and (**C**) TRP-2 at the transcription levels were examined by real-time PCR. (**D**) Tyrosinase, TRP-1, and TRP-2 at the translation levels were investigated by western blot. * *p* < 0.05 and ** *p* < 0.01 (vs. α-MSH-stimulated group). Arbutin (A, 0.5 mg/mL) was used as a positive control. Data are expressed as mean ± SEM (*n* = 3).

**Figure 7 plants-10-01633-f007:**
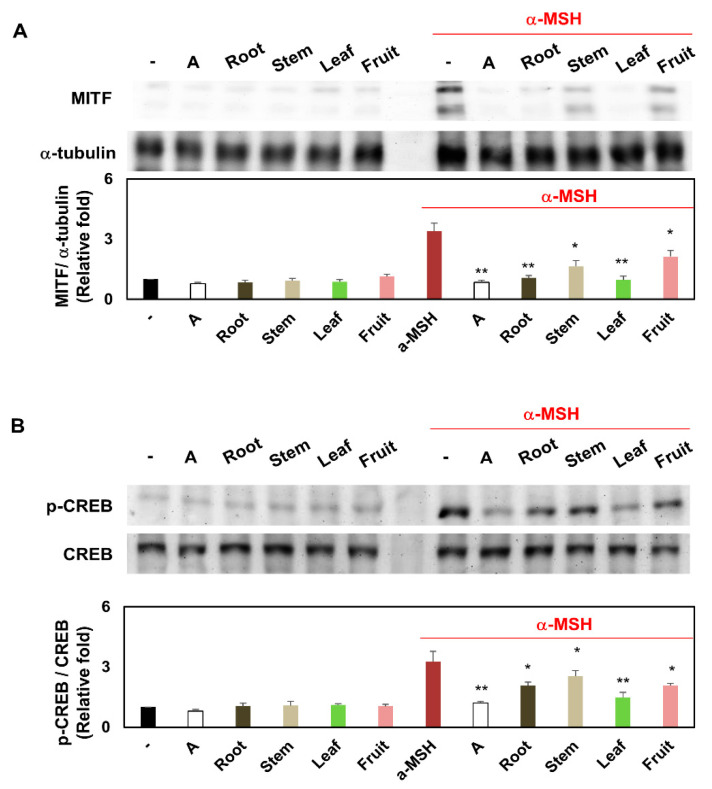
The MITF expression and CREB phosphorylation of KCR, KCS, KCL, and KCF extracts in α-MSH-stimulated melanocytes. Melanocytes B16F10 were treated with KCR, KCS, KCL and KCF extract (0.5 mg/mL) for 1 h after being treated to α-MSH (0.5 μM) or non-treatment. These Melanocytes were maintained for 48 h. (**A**) MITF at protein expression level was studied by western blot. (**B**) CREB at phosphorylation level was tested by western blot. * *p* < 0.05 and ** *p* < 0.01 (vs. α-MSH-stimulated group). Arbutin (A, 0.5 mg/mL) was used as a positive control. Data are expressed as mean ± SEM (*n* = 3).

## Data Availability

The data that support the findings of this study are available from the corresponding author upon reasonable request.

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
