# Peer review of "A Comparative Study on Photo-Protective and Anti-Melanogenic Properties of Different *Kadsura coccinea* Extracts"

_plants, 2021, doi:10.3390/plants10081633_

Round 1
Reviewer 1 Report
Review
Abstract
Full description of abbreviation α-MSH is missing.
In the sentence “KC extracts significantly inhibited the intracellular melanin content”, the word “inhibition” is not suitable since it is more used for processes, thant for a content.
“Tyrosinase” should start with lowercase, abbreviation TRP is not explained.
In the last sentence, the term “whitening agents” should be supplemented whit word “skin” for clarity.
Introduction
Lines 60-61. “In recent decades, UV radiation has become a serious skin health concern and is still spreading dangerously worldwide.” – What is spreading – radiation, concern, or UV-induced diseases?
Line 70. “Stratum basales” should be changed to “Stratum basale”.
Line 78. Full form of CREB is not given.
Materials and Methods
Chapter 2.1
Plant age, number of plants used in the study is unclear, also preparation procedure of different plant tissues is not described, although harvest date is repeated twice. What was the initial extract concentration in DMSO?
Chapter 2.2.
“As described above [22], the total polyphenolic and flavonoid contents of KC other partial extracts were measured…”- where “above” and what are “other partial extracts”?
“Absorbance was analyzed at…” should be changed to “measured at…”.
Chapter 2.3.
The title should be changed to show what was really analysed. “KC other partial extracts” is used again. The volume of extract or the ratio with a reagent is not given. What is the reason of the use of optical density (OD) instead of absorbance in this chapter?
Why two methods, DPPH and ABTS, are used for the evaluation of radical scavenging activity?
Chapter 2.4.
The medium was changed, or cells were re-cultured every 2–3 days? Cell culture origin, growing state of cells (suspension or adhered), and manufacturer of medium components are not given.
Chapter 2.6.
The abbreviation LDH is not given in full.
The principle of CCK-8 test is not described.
CCK-8 solution is added to cell suspension, however, the preparation procedure of suspension, volume, and number of cells in the suspension are not given.
Chapter 2.7.
A new cell line (RPE) appears in the text, which wasn’t mentioned in Chapter “2.4 Cell culture”. CM-H2DCFDA concentration in the cell medium used for incubation, microscope and flow cytometer parameters for DCF fluorescence measurement are missing.
Chapter 2.8
The abbreviation of PI is not given in full. It is unclear, how the apoptosis was calculated – were necrotic cells also evaluated in apoptosis calculation? A short assay explanation would bring more clarity.
Chapter 2.9
First sentence needs correction (“Intracellular melanin content and tyrosinase activity assays were measured by explaining where the slightly modified procedure differed [23]”). Second sentence (“Melanocytes were treated with α-MSH in the KC other partial extracts…”) also rise a question – are α-MSH and KC extract applied simultaneously? Maybe “and” fits better here than “in”. Again, the concentration of the extract and α-MSH in the cell medium is unclear. Is 0.5 mg/ml the final concentration in the medium or the initial extract concentration?
How formed dopachrome was distinguished from melanin since both were measured at 475 nm?
Chapter 2.10.
“GAPDH was used as an internal reference for Tyrosinase, TRP-1, and TRP-2” – incorrect, enzymes can not be used as reference in RNA/DNA method; unnecessary capitalization of “tyrosinase”. Why were TRP-1 and TRP-2 studied?
Results
Chapter 3.1.
“ABTS and DPPH activities” should be changed to “ABTS and DPPH radical scavenging activities”.
Chapter 3.2
The title and titles of further chapters should be revised and be of the same style, while now they are not uniform (“…under UVA…”, “…in the presence of UVA…”) and in some sense misleading since “under” or “in the presence” can be understood as simultaneous treatment with extracts.
One can understand the flow of the treatment only from Fig. 2 description - first UV, then treatment with the extract! But it must be clearly explained earlier in the methods and at the beginning of Chapter 3.2.
Fig. 2 description (“vs. UVA-irradiated group) should be corrected to “vs. UVA treatment only” or similar because all extract-treated groups were also UVA-irradiated groups.
Lines 240-241. Authors state that “…the antiproliferative and cytotoxic effects of UVA or UVB irradiation on keratinocytes were alleviated in the order of KCL, KCR, KCS, and KCF”, however, it would be also interesting to know whether any extract (maybe KCL) is able to restore cell viability to the control level.
Chapter 3.3.
In the first sentence, ROS are considered as “potential mediators of UVA or UVB irradiation”. Probably the meaning is “potential mediators of UVA or UVB radiation-induced damage”?
Sentence in lines 261-262 makes no sense: “keratinocytes repressed …staining in …irradiation”. Next sentence is also unclear – how the percentage of control (5.7 ± 0.2%) was obtained? What data were used for normalization (100%) in Fig. 3, 4 if “% of control” is shown on y axis and what was a control?
Figure titles must be revised since at a given form (“ROS production, apoptosis, etc. of …extracts…”) they are not acceptable. Also, unnecessary capitalization of words Melanocytes, Tyrosinase, etc. appears in the text.
C-D panels are not explained In Fig. 6.
All figures seem to be a little bit squeezed and lost its aspect ratio.
Discussion
The first sentence is unclear – what authors mean by “environmental pollution associated with UV irradiation” and “skin bleaching …is likely to reach high proportions in the coming decades for esthetic purposes” – high proportions of what and what are other purposes of skin bleaching than esthetic?
One term should be chosen instead of two terms used: “whitening candidates” and “blanching candidates”.
The sentence in lines 395-397 claim that fruits of KC are beneficial to human health because they are mostly consumed in the form of fresh fruits, juices, and fruit wine, what is not an argument.
The statement “The whitening candidates from natural extracts are non-toxic and can be safely used in cosmetic and pharmaceutical formulations to help develop natural whitening agents” is too general and can be wrong for numerous natural extracts.
The information provided in the discussion section on the role of MITF and CREB in melanogenesis would be more appropriate in the introduction to support the study.
The phrase “consistent with previous results” (line 411 and 432) needs reference.
The sentence “…this study is still limited to cell and animal experiments” is not correct since animals were not used in this study.
The last sentence seems to be unnecessary.
Author Response
Reviewer 1
Thank you for reviewing and providing your comments on my manuscript.
I have revised my manuscript in accordance with the comments from the Reviewer, and my point-by-point responses are listed below.
Abstract
Full description of abbreviation α-MSH is missing.
=> In accordance with the Reviewer’s comment, the abbreviation of α-MSH has been inserted.
[we aimed to explore the effects, functions, and mechanisms of the extracts from KC root (KCR), stem (KCS), leaf (KCL), and fruit (KCF) in UVA- and UVB-irradiated keratinocytes and α-melanocyte stimulating hormone (α-MSH)-stimulated melanocytes.]
In the sentence “KC extracts significantly inhibited the intracellular melanin content”, the word “inhibition” is not suitable since it is more used for processes, thant for a content.
=> In accordance with the Reviewer’s comment, I have corrected this sentence.
[Moreover, KC extracts significantly decreased the intracellular melanin content and tyrosinase activity in a-MSH-stimulated melanocytes.]
“Tyrosinase” should start with lowercase, abbreviation TRP is not explained.
=> In accordance with the Reviewer’s comment, I have corrected this sentence.
[Mechanistically, KC extracts reduced the protein and mRNA expression levels of tyrosinase, tyrosinase-related protein-1 (TRP-1), and tyrosinase-related protein-2 (TRP-2) in a-MSH-stimulated melanocytes.]
In the last sentence, the term “whitening agents” should be supplemented whit word “skin” for clarity.
=> In accordance with the Reviewer’s comment, I have corrected this sentence.
[Overall, the KC extracts exert photoprotective and anti-melanogenic effects, providing a basis for developing potential skin-whitening and photoprotective agents.]
Introduction
Lines 60-61. “In recent decades, UV radiation has become a serious skin health concern and is still spreading dangerously worldwide.” – What is spreading – radiation, concern, or UV-induced diseases?
=> It means UV induced diseases.
Line 70. “Stratum basales” should be changed to “Stratum basale”.
=> In accordance with the Reviewer’s comment, I have corrected this sentence.
[Melanocytes are distributed in the stratum basale of the skin epidermis, which is also affected by sunlight exposure, reactive oxygen species (ROS), and α-melanocyte stimulating hormone (α-MSH)]
Line 78. Full form of CREB is not given.
=> In accordance with the Reviewer’s comment, I have corrected this sentence.
[In addition, the myophthalmosis-related transcription factor (MITF) and cAMP responsive element binding protein (CREB) are transcription factors that primarily regulate melanogenesis and encode information about the mode and intensity of stimulation 1,2]
Materials and Methods
Chapter 2.1
Plant age, number of plants used in the study is unclear, also preparation procedure of different plant tissues is not described, although harvest date is repeated twice. What was the initial extract concentration in DMSO?
=> In accordance with the Reviewer’s comment, I have corrected this part.
[A 3-year-old KC 15 plant was grown in a 9 L plastic pot at Miryang campus of the Pusan National University. KC was identified by Professor Young Whan Choi, an author of this study. These samples were deposited as voucher specimens (accession number KC-PDRL-1) at the herbarium of the Department of Horticultural Bioscience, College of Natural Resources and Life Science, Pusan National University. Plants were watered suf-ficiently using a complete nutrient solution with a conductivity level of 1.0 mS·cm-1 and containing the following elements (in me∙L-1): NO3-N, 16; NH4-N, 1.34; P, 4; K, 8; Ca, 8; S, 4. KC gown in the port was harvested in December 2020 by classifying roots, stems, leaves and fruits (Fig. 1A). Harvested samples were immediately lyophilized in a freeze dryer and stored in vinyl bags at −20 °C until analysis. The dried roots, stems, leaves, and fruits of KC (20 g) were ground to a fine powder and extracted at room temperature with ethyl alcohol. Briefly, filtration and evaporation of EtOH extracts of KC were performed under reduced pressure at 45 °C, followed by lyophilization. Finally, the solid extract (50 mg/ml) was dissolved in dimethyl sulfoxide (DMSO) for further experiments.]
Chapter 2.2.
“As described above [22], the total polyphenolic and flavonoid contents of KC other partial extracts were measured…”- where “above” and what are “other partial extracts”?
=> In accordance with the Reviewer’s comment, I have corrected this sentence.
[As described above 3, the total polyphenolic and flavonoid contents of KC root, stem, leaf, and fruit extracts were measured using the Folin–Ciocalteu (total polyphenol) and aluminum chloride (flavonoid) colorimetric methods.]
“Absorbance was analyzed at…” should be changed to “measured at…”.
=> In accordance with the Reviewer’s comment, I have corrected this sentence.
[Absorbance was measured at 700 nm (total polyphenol) using Ultrospec 6300 Pro]
Chapter 2.3.
The title should be changed to show what was really analysed.
=> In accordance with the Reviewer’s comment, I have corrected this title.
[2.3. DPPH and ABTS assay]
“KC other partial extracts” is used again.
=> In accordance with the Reviewer’s comment, I have corrected this sentence.
[The DPPH and ABTS radical scavenging activities of KC root, stem, leaf, and fruit extracts (0.5 mg/mL) were measured according to a previously described method 3]
The volume of extract or the ratio with a reagent is not given. What is the reason of the use of optical density (OD) instead of absorbance in this chapter?
=> In accordance with the Reviewer’s comment, I have corrected this part.
[KC root, stem, leaf, and fruit extracts (0.5 mg/mL) were mixed with DPPH solution (60 µM) or ABTS solution (7 mM) in microplates. After the samples were shaken vigorously, they were kept in the dark at 25 °C for 0.5 hours. The Absorbance of the sample mixtures was monitored at 510 nm (DPPH) or 734 nm (ABTS).]
Why two methods, DPPH and ABTS, are used for the evaluation of radical scavenging activity?
=> In recent years, a wide range of spectrophotometric methods have been adopted to measure the antioxidant capacity of natural extracts, the most widely used being the ABTS and DPPH assays.The ABTS assay in the specification is based on the production of blue/green ABTS·+, which can be reduced by antioxidants. The DPPH assay, on the other hand, is based on the reduction of purple DPPH· to 1,1-diphenyl-2-picryl hydrazine.
Chapter 2.4.
The medium was changed, or cells were re-cultured every 2–3 days? Cell culture origin, growing state of cells (suspension or adhered), and manufacturer of medium components are not given.
=> In accordance with the Reviewer’s comment, I have corrected this sentence.
[The adhered Keratinocytes (HaCaT) and adhered melanocytes (B16F10) were inocu-lated in DMEM (Invitrogen Corporation, Carlsbad, CA, USA) culture solution containing 10% FBS (Invitrogen Corporation, Carlsbad, CA, USA) and 1% penicillin/streptomycin (Invitrogen Corporation, Carlsbad, CA, USA) and cultured at 37 °C with 5% CO2. The ad-hered HaCaT and B16F10 cell growth was observed regularly, and the medium was changed every 2–3 days. Cells from the logarithmic growth phase were used for subse-quent experiments.]
Chapter 2.6.
The abbreviation LDH is not given in full.
=> In accordance with the Reviewer’s comment, I have corrected this sentence.
[lactate dehydrogenase (LDH)]
The principle of CCK-8 test is not described.
CCK-8 solution is added to cell suspension, however, the preparation procedure of suspension, volume, and number of cells in the suspension are not given.
=> In accordance with the Reviewer’s comment, I have corrected this part.
[For cell viability analysis, CCK-8 solution from the CCK-8 Assay Kit (Sigma-Aldrich, St. Louis, MO, USA) was added to HaCaT keratinocytes and B16F10 melanoma cell sus-pensions according to the manufacturer's instructions and incubated at 37 °C for 4 h. Briefly, 2 x 104 cells were seeded into each well of a 24-well plate and incubated with 5% CO at 37 °C for 24 h. Briefly, 2 x 104 cells were seeded into each well of a 24-well plate and incubated with 5% CO at 37 °C for 24 h. After 24 hours of incubation, CCK-8 reagent was added to each well, and the cells were further incubated for 4 hours. A cytotoxicity detec-tion kit (Roche Applied Science, Switzerland) was used to determine extracellular lactate dehydrogenase (LDH) release in HaCaT keratinocyte culture medium. Absorbance was analyzed at 450 nm (CCK-8) and 490 nm (LDH) using a VICTOR Multilabel Plate Reader.]
Chapter 2.7.
A new cell line (RPE) appears in the text, which wasn’t mentioned in Chapter “2.4 Cell culture”. CM-H2DCFDA concentration in the cell medium used for incubation, microscope and flow cytometer parameters for DCF fluorescence measurement are missing.
=> In accordance with the Reviewer’s comment, I have corrected this part.
Intracellular ROS levels were analyzed using 5-(and-6)-chloromethyl-2',7'-dichlorodihydrofluorescein diacetate acetyl ester (CM-H2DCFDA; Thermo Fisher Scientific, Waltham, MA, USA). All procedures were per-formed according to the manufacturer’s instructions. In brief, after KC other partial ex-tracts (0.5 mg/mL) treatment, Keratinocytes (HaCaT) were washed with phos-phate-buffered saline (PBS) and incubated with CM-H2DCFDA (5 µM) for 0.5 h in the dark. Thereafter, the generation of intracellular ROS was visualized using a Carl Zeiss fluores-cence microscope; fluorescence intensity was measured based on a fluorescent dye (CM-H2DCFDA) using a flow cytometer (Fit NXT Flow Cyto, Thermo Fisher Scientific, Pasadena, CA, USA).
Chapter 2.8
The abbreviation of PI is not given in full. It is unclear, how the apoptosis was calculated – were necrotic cells also evaluated in apoptosis calculation? A short assay explanation would bring more clarity.
=> In accordance with the Reviewer’s comment, I have corrected this part.
[After exposure and treatment, HaCaT keratinocytes were trypsinized and centrifuged. Subsequently, apoptosis of the obtained cells was evaluated using the fluorescein isothio-cyanate (FITC) Annexin V/Dead Cell Apoptosis Kit (Invitrogen Life Technologies, Carls-bad, CA, USA) according to the manufacturer’s instructions. Briefly, keratinocytes were rinsed twice with PBS and keratinocytes in the annexin binding buffer were obtained and mixed with FITC/Annexin V (component A) and propidium iodide (PI) working solution. After incubation at room temperature for 15 min in the dark, keratinocyte apoptosis was measured and the percentage of apoptotic cells were calculated using a flow cytometer (Fit NXT Flow Cyto, Thermo Fisher Scientific, Pasadena, CA, USA). Signals were detected for FL1 (FITC/Nexin V) and FL3 (PI) channels, and quadrant marker staining and dotplots of stained cells were established.]
Chapter 2.9
First sentence needs correction (“Intracellular melanin content and tyrosinase activity assays were measured by explaining where the slightly modified procedure differed [23]”). Second sentence (“Melanocytes were treated with α-MSH in the KC other partial extracts…”) also rise a question – are α-MSH and KC extract applied simultaneously? Maybe “and” fits better here than “in”. Again, the concentration of the extract and α-MSH in the cell medium is unclear. Is 0.5 mg/ml the final concentration in the medium or the initial extract concentration?
=> In accordance with the Reviewer’s comment, I have corrected this part.
[Intracellular melanin content and tyrosinase activity assays were measured by ex-plaining where the slightly modified procedure differed [23]. Melanocytes were treated with final concentration of 0.5 μM a-MSH and the 0.5 mg/mL KC other partial extracts for 48 h. Melanocyte pellets were lysed with 1 N NaOH in 10% DMSO at 80 °C for 1 h. Rela-tive melanin content was determined by measuring absorbance at 475 nm using a VIC-TOR Multilabel Plate Reader. Intracellular tyrosinase activity was determined by measur-ing the rate of dopachrome production using L-DOPA. Melanocytes were washed with ice-cold PBS and lysed in PBS containing 1% (w/v) Triton X-100. The tyrosinase substrate L-DOPA (2 mg/mL) was prepared in the same phosphate lysis buffer. Each extract was placed in a 96-well plate, and enzyme analysis was initiated by adding L-DOPA solution. After incubation for 1 h, the absorbance was measured at 475 nm using a VICTOR Mul-tilabel Plate Reader to analyze dopachrome production. The value of each measurement was expressed as a percentage of the control. Arbutin (A, 0.5 mg/mL) was used as a posi-tive control.]
How formed dopachrome was distinguished from melanin since both were measured at 475 nm?
=> Dopachrome formation is based on spectrophotometric monitoring (wavelength of absorbance maximum of 475 nm) after oxidative cyclization of dopaquinone produced by tyrosinase-induced oxidation of substrates in the presence and absence of inhibitors.
Chapter 2.10.
“GAPDH was used as an internal reference for Tyrosinase, TRP-1, and TRP-2” – incorrect, enzymes can not be used as reference in RNA/DNA method; unnecessary capitalization of “tyrosinase”. Why were TRP-1 and TRP-2 studied?
=> In accordance with the Reviewer’s comment, I have corrected this sentence.
[GAPDH mRNA was used as an internal reference for tyrosinase, TRP-1, and TRP-2 mRNA.]
Results
Chapter 3.1.
“ABTS and DPPH activities” should be changed to “ABTS and DPPH radical scavenging activities”.
=> In accordance with the Reviewer’s comment, I have corrected this sentence.
[The total polyphenol and flavonoid contents and ABTS and DPPH scavenging activities were compared to determine the potential effects of KCR, KCS, KCL, and KCF extracts on antioxidant capacity.]
Chapter 3.2
The title and titles of further chapters should be revised and be of the same style, while now they are not uniform (“…under UVA…”, “…in the presence of UVA…”) and in some sense misleading since “under” or “in the presence” can be understood as simultaneous treatment with extracts.
One can understand the flow of the treatment only from Fig. 2 description - first UV, then treatment with the extract! But it must be clearly explained earlier in the methods and at the beginning of
=> In accordance with the Reviewer’s comment, I have corrected this part.
[3.2. Comparison of viable and damaged keratinocytes treated with several partial extracts of KC in the presence of UVA, UVB, or non-irradiation.
We conducted the following experiments to explore the effects of KCR, KCS, KCL, and KCF on keratinocytes. First, all the extracts were applied to HaCaT keratinocytes in the presence of UVA, UVB, or non-irradiation. CCK-8 analysis showed that the extracts did not significantly change keratinocyte viability at concentrations of 0.5 mg/ml. Later, keratinocytes were treated with KCR, KCS, KCL, and KCF in the presence UVA or UVB ir-radiation. UVA and UVB irradiation significantly inhibited keratinocyte viability, as shown by CCK-8 analysis in Figure 2A.]
Chapter 3.2.
Fig. 2 description (“vs. UVA-irradiated group) should be corrected to “vs. UVA treatment only” or similar because all extract-treated groups were also UVA-irradiated groups.
=> In accordance with the Reviewer’s comment, I have corrected this part.
Figure 2. The keratinocytes viability and cytotoxicity effects of KCR, KCS, KCL and KCF extract in the presence UVA, UVB or non-irradiation. Keratinocyte HaCaT were treated with KCR, KCS, KCL and KCF extract (0.5 mg/mL) for 1 h after being exposed to UVA (20 J/cm2), UVB (50 mJ/cm2) or non-irradiation. These keratinocytes were maintained for 24 h. (A) The percentage of viable keratinocytes was assessed using the CCK-8 assay. (B) The percentage of cytotoxicity was measured using the LDH assay. * p < 0.05 and ** p < 0.01 (vs. UVA-treatment only). +p < 0.05 and ++ p < 0.01 (vs. UVB-treatment only). Data are expressed as mean ± SEM (n = 3).
Lines 240-241. Authors state that “…the antiproliferative and cytotoxic effects of UVA or UVB irradiation on keratinocytes were alleviated in the order of KCL, KCR, KCS, and KCF”, however, it would be also interesting to know whether any extract (maybe KCL) is able to restore cell viability to the control level.
=> In accordance with the Reviewer’s comment, I have corrected this sentence.
[The above experimental results showed that the anti-proliferative and cytotoxic effects of UVA or UVB irradiation on keratinocytes were alleviated in the order of KCL, KCR, KCS, and KCF. Notably, KCL can restore cell viability to control levels.]
Chapter 3.3.
In the first sentence, ROS are considered as “potential mediators of UVA or UVB irradiation”. Probably the meaning is “potential mediators of UVA or UVB radiation-induced damage”?
=> In accordance with the Reviewer’s comment, I have corrected this sentence.
[As intracellular production of ROS causes severe keratinocyte damage, they are considered potential mediators of UVA or UVB radiation-induced damage.]
Sentence in lines 261-262 makes no sense: “keratinocytes repressed …staining in …irradiation”. Next sentence is also unclear – how the percentage of control (5.7 ± 0.2%) was obtained? What data were used for normalization (100%) in Fig. 3, 4 if “% of control” is shown on y axis and what was a control?
=> In accordance with the Reviewer’s comment, I have corrected this part.
[As intracellular production of ROS causes severe keratinocyte damage, they are con-sidered potential mediators of UVA or UVB radiation-induced damage. Several studies have suggested that UVA or UVB irradiation induces endogenous ROS production [12, 14]. We aimed to determine if there was an increase in endogenous ROS levels in UVA-or UVB-irradiated keratinocytes. Accordingly, we analyzed the intracellular fluorescence in-tensity of the probe CM-H2DCFDA using fluorescence microscopy and flow cytometry. As a result of fluorescence microscopy, CM-H2DCFDA staining images showed slight stain-ing in control, KCR, KCS, KCL, and KCF-treated keratinocytes and significant staining in UVA-or UVB-irradiated keratinocytes (Fig. 3A). According to the quantified results of flow cytometry, UVA and UVB irradiation increased intracellular ROS levels in keratinocytes by 27.2 ± 4.5% and 34.1 ± 4.2%, respectively, compared with that in the control (5.7 ± 0.2%). In addition, similar to the fluorescence microscopy results, it was confirmed that the in-tracellular ROS level was suppressed by KC extracts in the order of KCL> KCR> KCS> KCF in the presence UVA or UVB irradiation (Fig. 3B). These results indicate that several partial extracts of KC significantly inhibited keratinocyte damage by reducing endogenous ROS levels.]
=> fluorescence intensity was measured based on a fluorescent dye (CM-H2DCFDA) using a flow cy-tometer (Fit NXT Flow Cyto, Thermo Fisher Scientific, Pasadena, CA, USA).
Figure titles must be revised since at a given form (“ROS production, apoptosis, etc. of …extracts…”) they are not acceptable. Also, unnecessary capitalization of words Melanocytes, Tyrosinase, etc. appears in the text.
=> In accordance with the Reviewer’s comment, I have corrected this part.
[Figure 3. Effect of KCR, KCS, KCL and KCF extract on the intracellular ROS production in UVA, UVB or non-irradiated keratinocytes.
Figure 4. Effect of KCR, KCS, KCL and KCF extract on the apoptosis in UVA, UVB or non-irradiated keratinocytes.]
[These melanocytes were maintained for 48 h. (A) The percentage of viable melanocytes was as-sessed using the CCK-8 assay. Intracellular melanin content (B) and tyrosinase activity assays (C)]
C-D panels are not explained In Fig. 6.
=> In accordance with the Reviewer’s comment, I have corrected this part.
[(A) Tyrosinase, (B) TRP-1, and (C) TRP-2 at the transcription levels were examined by real-time PCR. (D) Tyrosinase, TRP-1, and TRP-2 at the translation levels were investigated by western blot.]
All figures seem to be a little bit squeezed and lost its aspect ratio.
=> In accordance with the Reviewer’s comment, I have corrected this part.
Discussion
The first sentence is unclear – what authors mean by “environmental pollution associated with UV irradiation” and “skin bleaching …is likely to reach high proportions in the coming decades for esthetic purposes” – high proportions of what and what are other purposes of skin bleaching than esthetic?
=> In accordance with the Reviewer’s comment, I have corrected this sentence.
[The popularity of skin whitening is increasing worldwide owing to the increase in UV irradiation, and it is likely to reach high proportions in the coming decades for esthetic purposes]
One term should be chosen instead of two terms used: “whitening candidates” and “blanching candidates”.
=> In accordance with the Reviewer’s comment, I have corrected this sentence.
[It is well known that these properties of whitening candidates are indispensable contributors to the cosmetic and pharmaceutical research and development]
The sentence in lines 395-397 claim that fruits of KC are beneficial to human health because they are mostly consumed in the form of fresh fruits, juices, and fruit wine, what is not an argument.
=> According to the literature review, there have been no reports of any relevant argument to date.
The statement “The whitening candidates from natural extracts are non-toxic and can be safely used in cosmetic and pharmaceutical formulations to help develop natural whitening agents” is too general and can be wrong for numerous natural extracts.
=> In accordance with the Reviewer’s comment, I have deleted this sentence.
The information provided in the discussion section on the role of MITF and CREB in melanogenesis would be more appropriate in the introduction to support the study.
=> In accordance with the Reviewer’s comment, I have corrected this sentence.
Multiple studies have stated that MITF and CREB pathways regulate melanogenesis. MITF is a key factor in the transcription of melanogenesis-related enzymes and the central regulator of melanogenesis. a-MSH leads to the expression of MITF through a signaling mechanism involving cAMP-related binding protein (CREB), then MITF enters the nucleus with the M-box sequence (AGTCATGTGCT) to promote the transcription of specific mel-anogenic genes and enzymes. It is also known that phosphorylated CREB is stimulated by a-MSH, which binds to the CRE consensus element in the Mitf promoter to upregulate Mitf transcription [21, 23, 33, 34]. => Insert introduction
The phrase “consistent with previous results” (line 411 and 432) needs reference.
=> In accordance with the Reviewer’s comment, I have corrected this sentence.
[Consistent with photoprotective results, the anti-melanogenic effects of KC leaves and roots were much higher than those of stem and fruit.]
The sentence “…this study is still limited to cell and animal experiments” is not correct since animals were not used in this study.
The last sentence seems to be unnecessary.
=> In accordance with the Reviewer’s comment, I have deleted this sentence.
Reviewer 2 Report
The manuscript presents a comparison of in vitro photo protective and anti-tyrosinase activities of various KC extracts. This work is very interesting, the experiments look well conducted and the manuscript is very readable. Although superior properties towards photoprotection anti-tyrosinase activity are demonstrated by KCL through in vitro cell culture studies, the commercial application needs dosage range identification in formulations through efficacy and safety studies. Efforts must be taken to benchmark polyphenol and flavonoid content of prospective KC extract and efficacy parameters as they are highly dependent on the geographical factors (plants sourced from different geographical areas).
Please check the following points to improve the quality of your article.
- Abstract-Please write a-MSH and LDH in full form when using for the first time
- Experimental
- DPPH and ABTS study:-
Please mention the positive and negative control used in the experiment
- UVA and UVB irradiation:-
What was the time of exposure to UVA and UVB? Please mention it to the experimental part to give more clarity
- Cytotoxicity:-
CCK-8write in full form when used for the first time
- Intracellular ROS:-
RPE cells are mentioned. Please clarify.
- Results
- Comparison of antioxidant activity and anti-tyrosinase activity to reference compounds is required
- Discussion
- Last paragraph-kindly check the formatting of this paragraph (various fonts and sizes are seen)
- The study focusses on in vitro cell culture studies. The term ‘animal experiments’ needs clarification.
Author Response
Reviewer 2
Thank you for reviewing and providing your comments on my manuscript.
I have revised my manuscript in accordance with the comments from the Reviewer, and my point-by-point responses are listed below.
The manuscript presents a comparison of in vitro photo protective and anti-tyrosinase activities of various KC extracts. This work is very interesting, the experiments look well conducted and the manuscript is very readable. Although superior properties towards photoprotection anti-tyrosinase activity are demonstrated by KCL through in vitro cell culture studies, the commercial application needs dosage range identification in formulations through efficacy and safety studies. Efforts must be taken to benchmark polyphenol and flavonoid content of prospective KC extract and efficacy parameters as they are highly dependent on the geographical factors (plants sourced from different geographical areas).
Please check the following points to improve the quality of your article.
Abstract-Please write a-MSH and LDH in full form when using for the first time.
=> In accordance with the Reviewer’s comment, the abbreviation of α-MSH and LDH has been inserted.
[α-melanocyte stimulating hormone (α-MSH), lactate dehydrogenase (LDH)]
Experimental
DPPH and ABTS study:-
Please mention the positive and negative control used in the experiment
=> In accordance with the Reviewer’s comment, I have corrected this part.
[DPPH radical scavenging activity of KCR, KCS, KCL and KCF extract (0.5 mg/mL). Data are expressed as mean ± SEM (n = 3). DMSO was used as a negative control.]
In this assay, kojic acid was used as a positive control, but data were not shown.
UVA and UVB irradiation:-
What was the time of exposure to UVA and UVB? Please mention it to the experimental part to give more clarity.
=> In accordance with the Reviewer’s comment, I have corrected this part.
[Keratinocytes were exposed to UVA or UVB irradiation (Bio-Link BLX-365; Vill-ber-Lourmat, Eberhardzell, Germany) with 5 × 8 W tubes that emit most of their energy at an emission peak at 365 nm (UVA) or 312 nm (UVB). UVA irradiation doses were 20 J/cm2 and UVB irradiation doses were 50 mJ/cm2.]
Cytotoxicity:-
CCK-8write in full form when used for the first time
=> In accordance with the Reviewer’s comment, the abbreviation of CCK-8 has been inserted.
Cell Counting Kit-8 (CCK-8)
Intracellular ROS:-
RPE cells are mentioned. Please clarify.
=> In accordance with the Reviewer’s comment, I have corrected this part.
Intracellular ROS levels were analyzed using 5-(and-6)-chloromethyl-2',7'-dichlorodihydrofluorescein diacetate acetyl ester (CM-H2DCFDA; Thermo Fisher Scientific, Waltham, MA, USA). All procedures were per-formed according to the manufacturer’s instructions. In brief, after KC other partial ex-tracts (0.5 mg/mL) treatment, Keratinocytes (HaCaT) were washed with phos-phate-buffered saline (PBS) and incubated with CM-H2DCFDA (5 µM) for 0.5 h in the dark.
Results
Comparison of antioxidant activity and anti-tyrosinase activity to reference compounds is required
=> In accordance with the Reviewer’s comment, I have included relevant references.
Discussion
Last paragraph-kindly check the formatting of this paragraph (various fonts and sizes are seen)
The study focusses on in vitro cell culture studies. The term ‘animal experiments’ needs clarification.
=> In accordance with the Reviewer’s comment, I have deleted this sentence.
Reviewer 3 Report
The Manuscript entitled “A comparative study on photo-protective and anti-melanogenic properties of different Kadsura coccinea extracts”, reports new and relevant information about whole composition of phenols and flavonoids, as well as several bioactivities of Kadsura coccinea extracts.
The experimental assays have been well designed, described, performed and interpreted. The conclusions are supported by the experimental results.
The Manuscript is acceptable for publication with minor modifications, as detailed below.
MINOR MODIFICATIONS
Section 2.12 Statistical Analysis
-Please, indicate the software used for multivariate statistics (SPSS or equivalent), such as ANOVA with Dunn post-hoc test.
References
Some recent and relevant references must be included and properly commented in the Introduction and/or the Discussion sections.
- Liu, J.S., Qi, Y.D., Lai, H.W., Zhang, J., Jia, X.G., Liu, H.T., Zhang, B.G., and Xiao, P.G. (2014). Genus Kadsura, a good source with considerable characteristic chemical constituents and potential bioactivities. Phytomedicine 21, 1092-1097.
- Sritalahareuthai, V., Temviriyanukul, P., On-nom, N., Charoenkiatkul, S., and Suttisansanee, U. (2020). Phenolic Profiles, Antioxidant, and Inhibitory Activities of Kadsura heteroclita (Roxb.) Craib and Kadsura coccinea (Lem.) A.C. Sm. Foods 9, 17.
- Tasneem, S., Yang, Y.P., Liu, B., Choudhary, M.I., and Wang, W. (2021). Cytotoxicity of Schisandronic Acid from Kadsura coccinea by Activation of Caspase-3, Cleavage of poly-ADP Ribose Polymerase, and Reduction of Oxidative Stress. Rev Bras Farmacogn-Braz J Pharmacogn 31, 51-58.
- Tram, L.H., Huong, T.T., Thuy, L.T., Thong, N.V., Anh, N.T., Minh, N.H., Ha, T.T., Dung, D.A., Thao, N.P., Thuong, P.T., et al. (2021). A new triterpenoid from the stems of Kadsura coccinea with their antiproliferative activity. Nat Prod Res, 6.
- Yang, Y.P., Hussain, N., Zhang, L., Jia, Y.Z., Jian, Y.Q., Li, B., Choudhary, M.I., Rahman, A.U., and Wang, W. (2020). Kadsura coccinea: A rich source of structurally diverse and biologically important compounds. Chin Herb Med 12, 214-223.
Best regards.
Author Response
Reviewer 3
Thank you for reviewing and providing your comments on my manuscript.
I have revised my manuscript in accordance with the comments from the Reviewer, and my point-by-point responses are listed below.
The Manuscript entitled “A comparative study on photo-protective and anti-melanogenic properties of different Kadsura coccinea extracts”, reports new and relevant information about whole composition of phenols and flavonoids, as well as several bioactivities of Kadsura coccinea extracts.
The experimental assays have been well designed, described, performed and interpreted. The conclusions are supported by the experimental results.
The Manuscript is acceptable for publication with minor modifications, as detailed below.
MINOR MODIFICATIONS
Section 2.12 Statistical Analysis
-Please, indicate the software used for multivariate statistics (SPSS or equivalent), such as ANOVA with Dunn post-hoc test.
=> In accordance with the Reviewer’s comment, I have corrected this part.
All assays were independently repeated at least three times. All statistical parameters are presented as the mean ± standard error of the mean (SEM). Statistical analyses were performed using one-way analysis of variance (ANOVA), followed by Dunn’s post-hoc test. A value of p<0.01 or p<0.05 was considered significant.
References
Some recent and relevant references must be included and properly commented in the Introduction and/or the Discussion sections.
- Liu, J.S., Qi, Y.D., Lai, H.W., Zhang, J., Jia, X.G., Liu, H.T., Zhang, B.G., and Xiao, P.G. (2014). Genus Kadsura, a good source with considerable characteristic chemical constituents and potential bioactivities. Phytomedicine 21, 1092-1097.
- Sritalahareuthai, V., Temviriyanukul, P., On-nom, N., Charoenkiatkul, S., and Suttisansanee, U. (2020). Phenolic Profiles, Antioxidant, and Inhibitory Activities of Kadsura heteroclita (Roxb.) Craib and Kadsura coccinea (Lem.) A.C. Sm. Foods 9, 17.
- Tasneem, S., Yang, Y.P., Liu, B., Choudhary, M.I., and Wang, W. (2021). Cytotoxicity of Schisandronic Acid from Kadsura coccinea by Activation of Caspase-3, Cleavage of poly-ADP Ribose Polymerase, and Reduction of Oxidative Stress. Rev Bras Farmacogn-Braz J Pharmacogn 31, 51-58.
- Tram, L.H., Huong, T.T., Thuy, L.T., Thong, N.V., Anh, N.T., Minh, N.H., Ha, T.T., Dung, D.A., Thao, N.P., Thuong, P.T., et al. (2021). A new triterpenoid from the stems of Kadsura coccinea with their antiproliferative activity. Nat Prod Res, 6.
- Yang, Y.P., Hussain, N., Zhang, L., Jia, Y.Z., Jian, Y.Q., Li, B., Choudhary, M.I., Rahman, A.U., and Wang, W. (2020). Kadsura coccinea: A rich source of structurally diverse and biologically important compounds. Chin Herb Med 12, 214-223.
Best regards.
=> In accordance with the Reviewer’s comment, I have included relevant references and corrected in the Discussion sections.
[TDM has multi-component and multi-target properties and greatly improves human biological effectiveness and quality of life [29]. Modern phytochemical research shows that KC contains a variety of ingredients, with lignans and terpenoids being the main ingredients [30]. More than 202 compounds have been identified, including dibenzocyclooctadiene lignans, Spirobenzofuranoid dibenzocyclooctadiene lignans, Arylnaphthalene lignans, Kadlongilactone triterpenoids and sesquiterpenoids [31,32]. The dried roots, stems, and leaves of KC have a broad tradition of use in TDM to treat rheumatoid arthritis, duodenal ulcers, gastrointestinal disorders, and gynecological problems. The dried roots of KC, with actions of clearing heat and eliminating toxins, inducing diuresis for removing edema. Fruits of KC are mostly consumed in the form of fresh fruits, juices, and fruit wine, indicating that they are beneficial to human health [7,33,34]. Previous studies also have shown that it is rich in bioactive ingredients such as lignans, triterpenoids, flavonoids, phenolic acids, steroids, and amino acids, which have high nutritional and medicinal value [3,35]].
Reviewer 4 Report
Interesting study which may have practical implications.
Remarks:
What was the final concentration of DMSO in the cell culture medium when extracts dissolved in DMSO we added to the cells? Was the effect of DMSO alone measured?
Line 101: “As described above [22]”, why “above”?
Lines 108 and following: “of KC other partial extracts”?
Line 115: “ABTS solution (7 mM)”. It seems imprecise. I guess that 7 mM ABTS solution was first oxidized to form ABTS radicals, most probably with persulfate; the procedure should be described in a more detail;
Line 154: “were measured by explaining where the slightly modified procedure differed”, somewhat unclear statement, perhaps “as described in [23] with minor modifications indicated”;
Line 191: “Immunodetection was performed using tyrosinase (1:1000)” etc., I guess: with anti-tyrosinase antibodies (1:1000), etc. Monoclonal or polyclonal? What was the origin of primary and secondary antibodies?
Line 207: ”ABTS and DPPH activities”, perhaps: ” ”ABTS and DPPH radical scavenging activities”;
Line 222: “Antioxidant effects”, rather “Antioxidant properties”;
Line 327, 355 and 376: “Melanocytes”, should be “melanocytes”;
Figures 5 - 7. Why the effect of arbutin is not shown?
Author Response
Reviewer 4
Thank you for reviewing and providing your comments on my manuscript.
I have revised my manuscript in accordance with the comments from the Reviewer, and my point-by-point responses are listed below.
Interesting study which may have practical implications.
Remarks:
What was the final concentration of DMSO in the cell culture medium when extracts dissolved in DMSO we added to the cells? Was the effect of DMSO alone measured?
=> Final concentration f DMSO is 1%. DMSO was used as a negative control.
Line 101: “As described above [22]”, why “above”?
=> In accordance with the Reviewer’s comment, I have corrected this sentence.
[As described previously [22], the total polyphenolic and flavonoid contents of KC root, stem, leaf, and fruit extracts were measured using the Folin–Ciocalteu (total polyphenol) and aluminum chloride (flavonoid) colorimetric methods.]
Lines 108 and following: “of KC other partial extracts”?
=> In accordance with the Reviewer’s comment, I have corrected this sentence.
Standard curves were constructed using gallic acid (total polyphenol) and quercetin (flavonoid) as the standards, and the results were expressed as gallic acid equivalents per gram (GAE/g) of KC root, stem, leaf, and fruit extracts and quercetin equivalent per gram (QE/g) of KC root, stem, leaf, and fruit extracts, respectively.
Line 115: “ABTS solution (7 mM)”. It seems imprecise. I guess that 7 mM ABTS solution was first oxidized to form ABTS radicals, most probably with persulfate; the procedure should be described in a more detail;
=> In accordance with the Reviewer’s comment, I have corrected this part.
[Stock solutions of 7 mM ABTS and 2.6 mM potassium persulfate were prepared in dis-tilled water at room temperature in the dark for 18 hrs. KC root, stem, leaf, and fruit ex-tracts (0.5 mg/mL) were mixed with the working solution and then allowed to stand for 0.5 h at room temperature in the dark. The Absorbance of the sample mixtures was moni-tored at 510 nm (DPPH) or 734 nm (ABTS).
Line 154: “were measured by explaining where the slightly modified procedure differed”, somewhat unclear statement, perhaps “as described in [23] with minor modifications indicated”;]
Line 191: “Immunodetection was performed using tyrosinase (1:1000)” etc., I guess: with anti-tyrosinase antibodies (1:1000), etc. Monoclonal or polyclonal? What was the origin of primary and secondary antibodies?
=> In accordance with the Reviewer’s comment, I have corrected this sentence.
[The goat anti-rabbit (IgG) secondary antibody (1:5000, Cell Signaling Technology) was added to the membrane and incubated at room temperature for 1 h.]
Line 207: ”ABTS and DPPH activities”, perhaps: ” ”ABTS and DPPH radical scavenging activities”;
=> In accordance with the Reviewer’s comment, I have corrected this sentence.
[The total polyphenol and flavonoid contents and ABTS and DPPH scavenging activities were compared to determine the potential effects of KCR, KCS, KCL, and KCF extracts on antioxidant capacity.]
Line 222: “Antioxidant effects”, rather “Antioxidant properties”;
=> In accordance with the Reviewer’s comment, I have corrected this sentence.
[To further investigate the antioxidant properties of KCR, KCS, KCL, and KCF, ABTS and DPPH radical scavenging assays were performed.]
Line 327, 355 and 376: “Melanocytes”, should be “melanocytes”;
=> In accordance with the Reviewer’s comment, I have corrected this part.
Figures 5 - 7. Why the effect of arbutin is not shown?
=> Arbutin was used as a positive control in anti-melanogenic studies.
This manuscript is a resubmission of an earlier submission. The following is a list of the peer review reports and author responses from that submission.